# The Evaluation of Dental Anxiety in Primary School Children: A Cross-Sectional Study from Romania

**DOI:** 10.3390/children7100158

**Published:** 2020-10-02

**Authors:** Ramona Vlad, Anca Maria Pop, Peter Olah, Monica Monea

**Affiliations:** 1Department of Odontology and Oral Pathology, George Emil Palade University of Medicine, Pharmacy, Science, and Technology of Târgu Mureș, 540139 Tirgu Mures, Romania; ramona.vlad@umfst.ro (R.V.); monica.monea@umfst.ro (M.M.); 2Faculty of Medicine, George Emil Palade University of Medicine, Pharmacy, Science, and Technology of Târgu Mureș, 540139 Tirgu Mures, Romania; 3Department of Medical Informatics and Biostatistics, George Emil Palade University of Medicine, Pharmacy, Science, and Technology of Târgu Mureș, 540139 Tirgu Mures, Romania; peter.olah@umfst.ro

**Keywords:** dental anxiety, cortisol, saliva, child health

## Abstract

Current data report that high levels of dental anxiety in children have a negative impact on oral health. The aim of this study was to measure dental anxiety, based on the Abeer Children Dental Anxiety Scale (ACDAS) used as a self-reported measure and to correlate its values with the salivary cortisol levels. The study was conducted in 2019 and included 389 children aged 6–9 years old; evaluation of dental anxiety and saliva sampling were performed. The influence of gender on the presence of dental anxiety was analyzed using Fisher’s exact test, the salivary cortisol level was compared between anxious and non-anxious children and was further correlated with the ACDAS score (*p* < 0.05). Girls had higher odds of experiencing dental anxiety (odds ratio: 1.533, *p* = 0.041). Salivary cortisol levels were higher in anxious compared to non-anxious children (median 1.251 vs. 1.091 ng/mL, *p* < 0.001) and showed a positive moderate correlation with the ACDAS score (*r* = 0.411, *p* < 0.001). Children aged 6–9 years have a high prevalence of dental anxiety, with girls being more susceptible to this condition. Salivary cortisol levels are higher in anxious children and correlate positively with the ACDAS score, proving that ACDAS can be used for the detection of dental anxiety.

## 1. Introduction

Dental anxiety has been recognized as a common condition that develops mostly during childhood and adolescence, being reported in 20–50% of cases. It is considered a behavioral disturbance with a variable nature, ranging from fears or phobias related to dental stimuli, such as needles or drilling, to a generalized anxiety frequently associated with the dental clinic environment [1,2]. Furthermore, children with high levels of dental anxiety were reported to have more decayed, missing and filled teeth, which led to the recognition of this condition as a negative factor with major implications on the oral health status. For dental professionals, it is difficult to differentiate between phobia (a marked fear that interferes with the normal routine of a person) and fear (which is not always extreme), and therefore these conditions are addressed to as dental anxiety [3,4].

For a long time, the study of dental anxiety was based exclusively on questionnaires, considered to be the most reliable methods to assess anxiety in children with cognitive ability to self-report their feelings on a scale. The evaluation of an anxiety state based on objective measures is very important, as clinical observations alone proved to be unreliable, with poor to moderate agreement between the dentist’s and child’s own evaluation of anxiety status. Children are often accompanied by parents to the dental clinic and data from the literature reported that the pediatric specialist conducts most of the communication with the parent, rather than focusing on the young patient [5].

Despite the great number of self-reported measures available, none of them could be regarded as ideal and suitable to evaluate children’s dental anxiety worldwide [6,7,8]. Hence, there was a need to develop a new method for the assessment of dental anxiety by using other factors that contribute to dental anxiety, such as the cognitive, behavioral, and psychological features. As a consequence, in the attempt to overcome all the shortcomings of previously used scales, Abeer Al-Namankany et al. [9] introduced in 2012 a new dental anxiety scale that was suitable for children and adolescents, known as Abeer Children Dental Anxiety Scale (ACDAS).

The quantification of dental anxiety is best obtained by combining psychological methods, such as questionnaires, with physiological variables which are objective measures. Various parameters, such as heart rate, nervous and muscular activity or palmar sweating, have been used as indicators of dental anxiety in children [10]. Moreover, a positive correlation was found between heart rate, salivary cortisol and the results of self-reported measures [11]. Stress induced by dental procedures was associated with an increased release of cortisol, a glucocorticoid which is transferred from blood to saliva within no more than 2–3 min. Data from the scientific literature confirmed that the levels of cortisol could offer accurate and reliable information regarding the level of stress in both children and adults and therefore it was used in dental research for measuring the levels of stress in relation to dental treatments [12]. Furthermore, by using saliva, the discomfort due to the need for a venous puncture that might generate false positive results is eliminated and therefore, the salivary levels of cortisol have been proposed as a biomarker for stress. 

The experience of anxiety during dental treatments usually prevents patients from fully cooperating with dental specialists, is accompanied by an increased time commitment from the practitioner, difficulties during treatment visits and also has a negative influence on the effectiveness of the dental treatment, delaying the detection of oral and dental diseases [13]. Dental anxiety is a widespread problem, affecting patients of all ages, as the methods of treatment are usually invasive [14]. Therefore, dental professionals must take into consideration the numerous etiological factors of anxiety, for a better understanding of the reactions of their young patients, as this emotional experience prevents children from receiving the dental treatment they need, with negative consequences on their oral health [13,14].

In our country, data regarding children’s anxiety in relation to dental treatment have gained increasing interest over the last few years, but until now, few data have been reported, based only on questionnaires. Therefore, the purpose of our study was to evaluate the prevalence of dental anxiety in a group of children between 6 and 9 years of age, before a routine dental control, using the ACDAS score as self-reported measure and the levels of salivary cortisol as a biological marker of stress; furthermore, we evaluated the correlations between these two variables. The null hypotheses to be tested are the following: (1) gender has no influence on the presence of dental anxiety in children, (2) there is no statistically significant difference between the salivary cortisol levels in anxious and non-anxious children, respectively, (3) the ACDAS scores do not correlate with the salivary cortisol values.

## 2. Materials and Methods

### 2.1. Study Design and Participants

This cross-sectional study was conducted between March and June 2019 in a group of 389 children of 6–9 years old (188 boys and 201 girls) from Târgu Mureș, as part of a protocol between our faculty and three primary schools, regarding the application of dental preventive measures. As inclusion criteria we used the following: children aged 6–9 years old, with no learning disabilities, who had attended a dental office before and agreed to cooperate with the examiner. We excluded from the study children whose parents did not approve their enrolment in this research, children with asthma (the use of inhaled corticosteroid influences the hypothalamic–pituitary–adrenal axis, reducing the level of endogenous cortisol) and those with gingival bleeding on the examination day (according to the manufacturer’s instructions, blood contamination of saliva samples should be avoided).

The sample size was calculated based on the following formula recommended for prevalence studies: n $=\frac{Z^{2}P\left(1-P\right)}{d^{2}}$, considering n = the sample size, Z = the statistic for a level of confidence, P = the expected prevalence and d = the precision (according to the effect size). The final sample size recommended for this study was 385, after using the following parameters: Z = 1.96 (corresponding to a level of confidence of 95%), P = 50% (conventional value chosen due to the lack of a clear prevalence in the literature) and d = 0.05 (appropriate for a prevalence between 10–90%). An excess of 10% was added in order to overcome number reduction due to drop out or the loss of participants over the course of the study [15,16]. (Figure 1)

### 2.2. Subjective and Objective Measures

We used the dental part of ACDAS, which contains 13 questions with 3 possible answers based on a Likert scale using faces: 1—relaxed, not scared, 2—neutral, feeling OK, 3—scared or feeling anxious (Table 1). For each question the answer was recorded as 1, 2 or 3 (range 13–39) and a score < 26 meant that the child was not experiencing dental anxiety. 

The saliva samples were collected between 12 and 1 P.M. in order to avoid bias due to the diurnal variation of cortisol levels and consisted of approximately 2 mL saliva stored at −20 °C until evaluation. The measurements were conducted using the Cortisol Saliva ELISA Assay Kit (IBL International GmbH, Hamburg, Germany) according to the manufacturer’s instructions. The range of detection varied between 0.15 and 30 ng/mL.

### 2.3. Clinical Evaluation Protocol and Data Collection

Groups of 15 children were scheduled and accompanied by their teachers to the dental clinic, based on a pre-established timetable. The examinations were conducted in the Clinic of Odontology and Oral Pathology from our faculty by the same investigator (first author), so there was no need for inter-observer calibration. The first part was represented by the measurement of ACDAS score, each participant being asked to indicate the face that best represented her/his response to the questions. The second part consisted of saliva sampling. Prior to evaluation, children were asked not to drink, eat or chew gum one hour before saliva collection; inspection of the oral cavity was performed by a dental specialist (R.V.) in order to exclude patients with oral lesions which could contaminate samples with blood. The oral cavity was rinsed with water 5 min before saliva sampling. Each child was seated and asked to bend the head slightly forward and to avoid swallowing in order to passively collect the saliva inside the oral cavity. After 3 min, they were asked to let the saliva flow into a special sterile recipient and the procedure was repeated after approximately 2 mL of saliva were collected.

### 2.4. Statistical Analysis

The results were statistically analyzed using GraphPad Prism 8 for Windows (GraphPad Software, San Diego, CA, USA). The continuous variables were reported as mean ± standard deviation (SD) for normally distributed variables and as median and interquartile range (IQR) for non-normally distributed variables. Categorical variables were reported as absolute numbers and percentages. The contingency tables were analyzed using Fisher’s exact test and the degree of association between variables was approximated by the value of odds ratio (OR), with 95% confidence interval (CI). All continuous data were evaluated for normal distribution using the Kolmogorov–Smirnov test and were further assessed using Student’s t test or the Mann–Whitney test. The association between ACDAS score and salivary cortisol level was analyzed by Spearman correlation. The level of statistical significance was set at *p* < 0.05 (two-tailed). 

### 2.5. Ethics Statement

The study was conducted according to the Declaration of Helsinky and ethical approval for this investigation was previously obtained from the Ethics Committee of the George Emil Palade University of Medicine, Pharmacy, Science, and Technology of Târgu Mureș (No. 274/21.11.2018). Written consent was signed by the parents or legal representatives, and children were also asked if they agreed to participate in this study.

## 3. Results

The evaluation was based on the results obtained from 389 children, with a mean age of 7.57 ± 1.32 years. The girls had a slightly higher age than boys (7.62 ± 1.41 compared to 7.54 ± 1.28), but the difference was not statistically significant (*p* = 0.56). The distribution of the study group according to age and gender is presented in Table 2. 

The girls represented 51.67% of the whole group and the boys 48.33%. The prevalence of dental anxiety for the whole group, based on ACDAS scores, was 43.7% (25.2% girls and 18.5% boys). According to gender, the prevalence of anxiety was 48.75% for girls and 38.29% for boys. Based on ACDAS scores, the distribution of anxious/non-anxious children according to gender is presented in Table 3. The statistical analysis based on Fisher’s exact test reported an OR = 1.533, with a CI ranging between 1.024 and 2.295 and *p* = 0.041, which shows that girls have statistically significant higher odds of developing dental anxiety. 

The levels of salivary cortisol (ng/mL) expressed as median and IQR were 1.124 (0.893–1.354) for girls and 1.196 (0.933–1.409) for boys. The analysis using Mann–Whitney tests revealed no statistically significant difference between genders (*p* = 0.2). The levels of salivary cortisol according to anxiety status and gender are presented in Table 4.

The cortisol levels evaluated using Mann–Whitney tests were statistically significant higher in anxious compared to non-anxious girls (*p* = 0.014) and boys, respectively (*p* = 0.004). The same observation was noticed for the whole study group, with anxious participants having higher levels of cortisol (*p* < 0.001). 

The degree of association between the two variables (ACDAS score and salivary cortisol level) was assessed using Spearman correlation, with *r* = 0.411 (CI: 0.323–0.493) and was interpreted as a statistically significant moderate correlation (*p* < 0.001) (Figure 2).

## 4. Discussion

Dental anxiety is considered as starting during childhood and in the absence of diagnosis and management it has the tendency to increase with time. Scientific data have established that children who experience a high level of dental anxiety also have a greater number of untreated dental lesions [3,17,18]. A child’s dental fear is an important aspect for the dentist, as it might be linked to other behavioral problems, such as attention-deficit/hyperactivity disorder (ADHD); therefore, the specialist should be aware of the level of dental anxiety before starting the treatment [19,20,21].

In children, the prevalence of dental anxiety is influenced by the age of the subjects and the diagnostic method employed. Al-Namankany et al. [22] and DeMenezes Abreu et al. [23] questioned the ability of pediatric patients to accurately report their anxiety or fear; it was concluded that children over 6 years old are able to express reliably on all aspects of their health, having better abilities in understanding and reporting their emotional states, compared to younger children [22]. Compared to the Children’s Fear Survey Schedule–Dental Subscale (CFSS-DS), which has been one of the most widely used questionnaires [24], ACDAS can better differentiate anxiety and meets the requirements to be considered a gold standard. Comparative analysis showed a strong correlation between the two scales [9,25].

An important issue is the language complexity and terminology used in the questionnaires, as, for example, the word “worried” was understood better by more children aged 5–7 years old compared to “nervous”. Previous studies suggested that there is an inverse correlation between age and the level of dental anxiety, but this is a matter of intense debate in the literature [18,26,27,28]. 

The higher levels of anxiety reported in girls were confirmed by our study. The distribution of our sample according to gender was approximately equal, but girls had a statistically significant higher percentage of anxiety (48.75%) compared to boys (38.29%). Therefore, we considered that gender has an influence on the prevalence of anxiety defined by ACDAS scores. However, data from literature suggested that females report dental anxiety more reliably than males, who tend to hide their fear [29]. The age group chosen for the current study (6–9 years) can be considered as belonging to a homogenous population, based on the evidence provided by previous studies [30].

Our rationale for evaluating dental anxiety based on a questionnaire and a biological marker of stress before a non-invasive, prophylactic dental procedure is supported by the findings of Gomes et al. [31], who reported that even such procedures could be a stress trigger in young patients. Moreover, similarly to Furlan et al. [32], the authors concluded that the salivary cortisol levels are higher before dental prophylaxis than after. This observation reveals that the stress experienced by the child is of psychological nature, rather than interventional. In our study, the oral inspection performed by the dental specialist prior to saliva sampling could have been a trigger for dental anxiety, with a possible influence on the salivary cortisol levels.

Dental prophylactic measures may be considered a way of acclimatizing the child to the dental clinical environment, as dental anxiety is known to be more frequent in children who have never been scheduled for a dental appointment before. The high prevalence of anxiety in our study group could be related to the fact that all investigations were conducted in the clinic environment, prior to the routine dental control.

When saliva is compared to other bio-fluids, it becomes very attractive due to the non-invasive nature of sampling, low cost and greater clinical safety, which contributed to the idea that it represents “a window of opportunity” for modern medicine. A good example for this concept is the determination of salivary cortisol, enhanced by similar levels with serum cortisol and easy conservation (1–2 days at room temperature or 1 week at +4 °C) [33].

Salivary cortisol levels are not affected by the rate of salivary flow, being relatively resistant to enzymatic degradation. Furthermore, studies proved that there is a high correlation between serum and salivary cortisol, indicating that the latter can be used as a reliable estimate of serum cortisol levels. Cortisol production has a circadian rhythm, with levels peaking in the early morning and dropping to lowest values at night [31]. However, levels rise independently of circadian rhythm in response to stress. Unbound serum cortisol enters the saliva via intracellular mechanisms and equilibrium between serum and saliva is achieved in less than 5 min [34]. The use of saliva as diagnostic fluid overcomes the disadvantages of venous puncture, which might induce additional stress and therefore contribute to false positive results. Furthermore, salivary cortisol represents the active fraction of the hormone, not the total value, which is measured from serum [35].

ACDAS scores showed a positive moderate correlation with the salivary cortisol levels. There are also a series of factors which could have influenced the correlation coefficient: the range of values and especially the presence of outliers, which are valid values probably caused by the lack of understanding of the sampling protocol or questionnaire items by a participant. Our results show that ACDAS is a valid indicator of the dental anxiety status and can be used as a simple, cost effective tool by the dental practitioner.

Our study encountered several limitations, such as: the lack of recording of a caries index which could have given an appreciation of the oral health status and the lack of investigation regarding possible sources of dental anxiety (parents’ anxiety, previous history in the dental office). Future investigation is needed in order to measure the parents’ level of dental anxiety and evaluate if there is a correlation with the child’s anxiety status. However, the salivary cortisol level, despite being used as a biomarker of stress, is predisposed to being affected by various confounding factors and does not offer the possibility to differentiate fear in the dental clinic from chronic anxiety. Another limitation of our study is that we did not compare the salivary cortisol levels in clinical and non-clinical (classroom) settings. Moreover, a more precise evaluation of salivary cortisol levels could have been obtained if saliva samples were collected before answering the questionnaire items. Based on the results of our study, we found a high prevalence of dental anxiety among children aged 6–9 years, an aspect which may be important for pediatric dentists. It could be a starting point for the development of prophylactic strategies aiming to reduce the level of dental anxiety in children. Despite the limitations of our research, there is a definite association between dental anxiety quantified based on a subjective measure and salivary cortisol level, which may facilitate the use of ACDAS as a reliable tool in the detection of dental anxiety. 

## Figures and Tables

**Figure 1 children-07-00158-f001:**
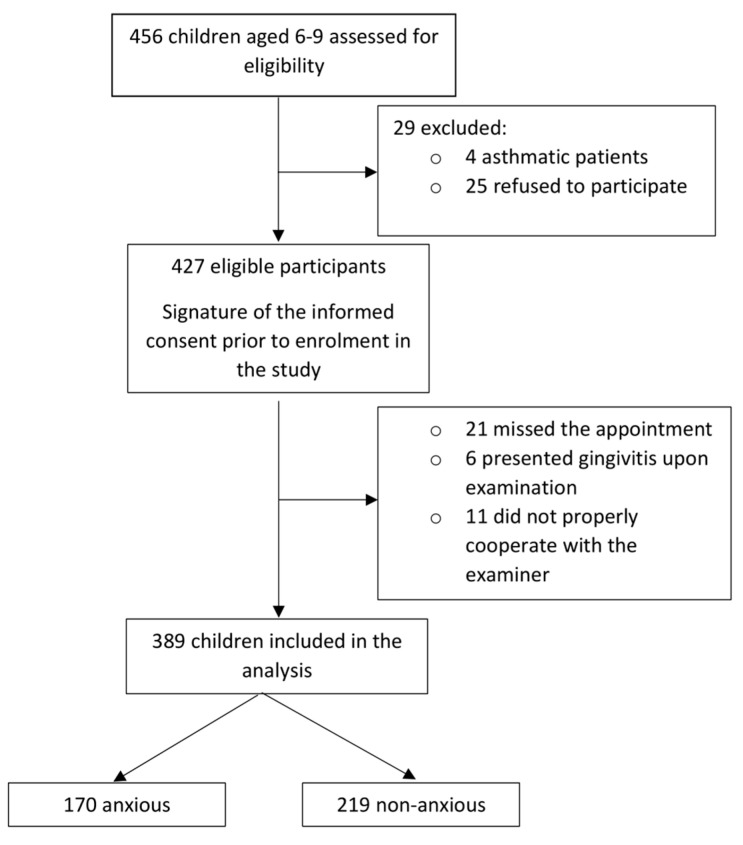
Flow diagram showing patients selection for the study.

**Figure 2 children-07-00158-f002:**
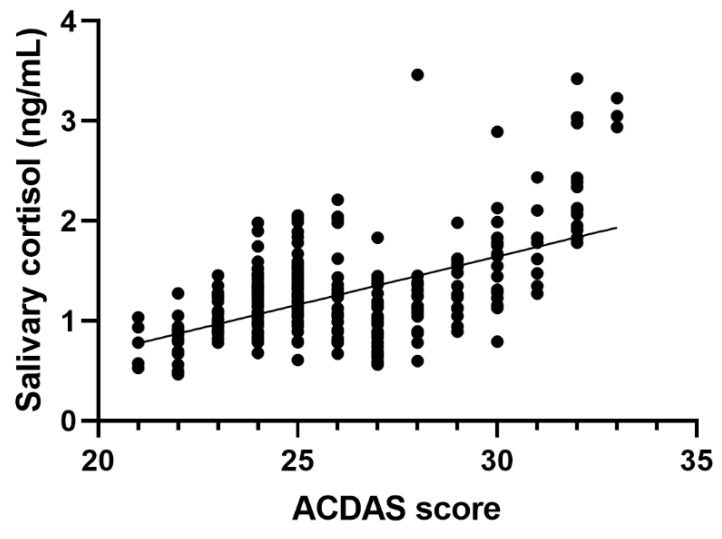
Correlation between ACDAS score and level of salivary cortisol (ng/mL).

**Table 1 children-07-00158-t001:** The dental part of the Abeer Children Dental Anxiety Scale (ACDAS).

How Do You Feel about	Happy (Score 1)	OK (Score 2)	Scared (Score 3)
1. Sitting in the waiting room?			
2. A dentist wearing a mask on his face?			
3. Laying flat on the dental chair?			
4. A dentist checking your teeth with a mirror?			
5. Having a strange taste in your mouth? (from filling material or gloves)			
6. Having a “pinch” feeling in your gum?			
7. The feeling of numbness (fat lip or tongue)?			
8. A dentist cleaning your teeth by buzzy electric arm that is spraying water?			
9. The sounds that you hear at the dentist?			
10. The smell at the dentist?			
11. Having a tooth taken off?			
12. Wearing a small rubbery mask on your nose to breath special gas to help you feel comfortable during treatment?			
13. Having a “pinch” feeling on the back of your hand?			

(Adapted from Al-Namankany et al., 2012 [9]).

**Table 2 children-07-00158-t002:** Distribution of the study group according to age and gender.

Years of Age	Girls n (%)	Boys n (%)	Total n (%)
**6.4–7.0**	38 (9.67%)	42 (10.75%)	80 (20.42%)
**7.1–8.0**	50 (12.92%)	47 (12.02%)	97 (24.94%)
**8.1–9.0**	54 (13.97%)	53 (13.62%)	107 (27.59%)
**9.1–9.3**	59 (15.11%)	46 (11.94%)	105 (27.05%)
**Total**	201 (51.67%)	188 (48.33%)	389 (100%)

**Table 3 children-07-00158-t003:** Distribution of dental anxiety according to gender, based on ACDAS scores (statistical analysis with Fisher’s exact test).

	Anxious (ACDAS ≥ 26)	Non-Anxious (ACDAS < 26)	*p* Value
**Girls**	98	103	0.041
**Boys**	72	116	

**Table 4 children-07-00158-t004:** Distribution of salivary cortisol levels (ng/mL) within the study group according to gender and anxiety status based on ACDAS score (statistical analysis with a Mann–Whitney test).

Salivary Cortisol Level (ng/mL)Median and IQR	*p* Value
	Anxious (ACDAS ≥ 26)	Non-Anxious (ACDAS < 26)	
Whole study group	1.251 (0.916–1.621)	1.091 (0.907–1.316)	<0.001
Girls	1.24 (0.89–1.794)	1.02 (0.903–1.275)	0.014
Boys	1.29 (1.03–1.619)	1.147 (0.907–1.35)	0.004
*p* value	0.5649	0.1251

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
