# Peer review of "The Evaluation of Dental Anxiety in Primary School Children: A Cross-Sectional Study from Romania"

_children, 2020, doi:10.3390/children7100158_

Round 1
Reviewer 1 Report
Dear Authors:
The paper topic is novel and I am pleased to see its exploration, thank you for your submission. There are some changes which I would recommend to improve the quality of the paper, and ensure the statements made are accurate.
Generally, the grammar can be improved and I would recommend that it is proof read prior to resubmission.
Introduction
Line 63: should 'de specialists' be dental specialists? needs altered
Line 63-64: Need references for the points made regarding the impact of dental anxiety on ability to cooperate with treatment, and the subsequent consequences for the dentist. Would suggest care be taken with use of the phrase 'waste of time' - I would argue that this phrase is not appropriate, you could consider revising this to something like 'increased time commitment from practioner'.
Line 66: Must provide reference for the view that all patients have a degree of dental anxiety.
Line 70: Reference required for the statement that dental anxiety negatively affects oral health
Materials and methods
Line 82: I am unfamiliar with the spring semester dates, please indicate the actual time period over which data was collected. (e.g. 3 months)
Line 88: please make clear to the reader here why children with chronic conditions were excluded, and why children with gingival bleeding were excluded.
Line 96: the 10% is said to be added to account for 'exclusion of participants from various reasons' - I think what you mean is to allow for participant drop out/loss over the course of the study. This should be mentioned regardless.
LINE 119: Who examined the oral cavity? This should be stated. If the clinician, does this not count as intervention, and would you consider that this could have influenced the subsequent cortisol level?
Results
Line 148: Is this proportion of anxious patients based on the questionnaire results, cortisol levels or both? Please clarify.
If based on questionnaire:
Line 154: Acknowledgement must be made that females are known to report dental anxiety more reliably than males, rather than being proof that females actually suffer greater dental anxiety than males.
Discussion
Line 178: What is meant by the term 'behavioural problems' - is this relating to conditions such as Autism/ADHD? A reference must be provided for this assertion if so.
Line 181: 'Researchers had been preoccupied by" - is this to indicate the previous focus on self reported measures? Consider rewording.
Line 186: Reference for CFS being most widely used dental fear scale.
Line 204: You appear to have used a reference for a paper here out of context, please revise the statement or remove the reference.
Author Response
Dear Reviewer,
Thank you for your appreciation regarding our research and for your valuable comments on our manuscript. In the attachment you may find our response to each of your observations.
Yours faithfully,
Anca Maria Pop

Reviewer 2 Report
The results of the study are very helpful for pediatric dental specialists. The design is perfectly structured. No marks about statistical analysis, well presented and illustrated. Discussion is a little bit confused at the beginning. There are several major points to reconsider:
- too long period of saliva collection after the moment of subjective evaluation of anxiety (It be more precise to evaluate the anxiety by the questionnaire exactly after the end of the collection of saliva)
- There is no information about the salivary cortisol levels in non-stressful situation. Is there a degree that presents state of anxiety or any detection of cortisol in saliva is interpreted as anxiety?
- There is no discussion and interpretation of the results regarding gender differences in anxiety levels. According to the questionnaire there is a difference, while as to cortisol level - no. Why?
- it will be better to cite other similar articles that compare subjective and objective measures of dental anxiety (for example heart rate, etc.)
Author Response

(The authors gave the same response as above.)

Round 2
Reviewer 2 Report
No recommendations.